# Structured Predictive Representations in Reinforcement Learning

## Abstract

Reinforcement Learning (RL) remains brittle in complex environments characterized by sparse rewards, partial observability, and subtask dependencies. Predictive state abstractions capture the environment's underlying temporal structure and are crucial to overcoming these challenges. Yet, such methods often only focus on global one-step transitions and overlook local relationships between trajectories. This paper explores how capturing such relationships can enhance representation learning methods in RL. Our primary contribution is to show that incorporating a Graph-Neural Network (GNN) into the observation-predictive learning process improves sample efficiency and robustness to changes in size and distractors. Through experiments on the MiniGrid suite, we demonstrate that our GNN-based approach outperforms typical models that use Multi-layer Perceptrons (MLPs) in sparse reward and partially-observable environments where task decompositions are critical. These results highlight the value of structural inductive biases for generalization and adaptability, revealing how such mechanisms can bolster performance in RL.

## 1 Introduction

Environments with partial observability, sparse rewards, and dynamic changes frequently challenge Deep Reinforcement Learning (RL) algorithms, often rendering them brittle and sample-inefficient (Wang et al., 2019; Meng & Khushi, 2019; Lu et al., 2020; Tomar et al., 2023; Benjamins et al., 2023). Traditional RL methods struggle particularly in such complex environments due to the challenges of capturing long-term dependencies and relational structures between states. Learning representations of the state relevant to control offers a promising avenue to scale RL to complex scenarios. *State abstractions* in Markov Decision Processes (MDPs) (Dayan, 1993; Dean & Givan, 1997; Li et al., 2006) and *history abstractions* in Partially Observable MDPs (POMDPs) (Littman et al., 2001; Castro et al., 2009) improve data efficiency and generalization (Killian et al., 2017; Zhang et al., 2021). Consequently, numerous RL representation learning techniques have emerged in the last years (Castro et al., 2021; Schwarzer et al., 2021; Hansen-Estruch et al., 2022; Lan & Agarwal, 2023; Guo et al., 2020; Grill et al., 2020) making it an active area of research in RL.

*Self-prediction* has positioned itself as a prominent technique for learning state abstractions. It is a self-supervised mechanism that uses a latent model to predict the next latent state using the current latent state and action as inputs (Guo et al., 2019; 2020; Grill et al., 2020; Schwarzer et al., 2021; Lee et al., 2021). In doing so, it approximates the one-step transition structure in the latent space (Tang et al., 2023; Voelcker et al., 2024; Khetarpal et al., 2024). This objective is also connected to the objective to predict subsequent observations in POMDPs (Ni et al., 2024), allowing the agent to approximate the actual transition dynamics in the belief space (Schrittwieser et al., 2020; Subramanian et al., 2022). Real-world environments, however, often come with rich local structure as well (Mohan et al., 2024), which is usually overlooked by these methods.

This paper investigates how leveraging Graph Neural Networks (GNNs) (Battaglia et al., 2018) within a self-predictive framework can enhance representation learning in RL in sparse reward and partially observable settings. Specifically, we propose a method that

captures relationships between a batch of latent states generated by a history encoder. This approach enables the model to encode temporal and relational dependencies in the observation-prediction mechanism, improving the sample's learning efficiency and robustness to environmental changes. In contrast to commonly used Multi-Layer Perceptron (MLP)-based methods, which often struggle with long-term dependencies and partial observability, GNNs excel at capturing relational structure between the latent states produced over time (see Figure 1.

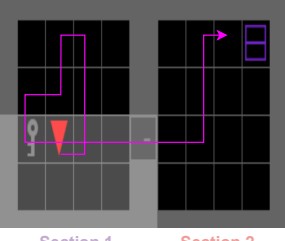 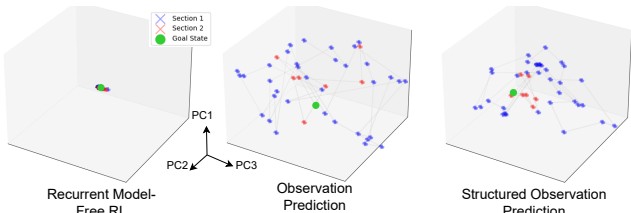

Section 1    Section 2

Recurrent Model-Free RL    Observation Prediction    Structured Observation Prediction

Figure 1: **Latent Space Representation.** A goal-reaching trajectory in `MiniGrid-UnlockPickup-v0` mapped to a 3D PCA representation of the latent states generated by various belief encoders. States belonging to the first section are indicated in blue, while those in the second one are shown in red, with the goal state highlighted in green. Structured Observation Prediction captures the closeness of high-reward states (red) near the goal. In contrast, normal Observation Prediction reveals a less organized representation, indicating potential inefficiencies in recognizing rewarding states in this environment. This emphasizes the advantage of graph-based approaches for improved decision-making and performance in reinforcement learning tasks.

This paper's main **contribution** is the introduction of a GNN-based observation-predictive model designed to operate on latent states generated by a history encoder. Unlike prior work that primarily focuses on spatial relationships (e.g., object-centric representations), our method targets temporal and relational dependencies in POMDPs. By relationally reasoning over trajectories, our method generalizes across variations in tasks. We validate our approach through experiments on a subset of navigation tasks in MiniGrid (Chevalier-Boisvert et al., 2023) that are particularly challenging for end-to-end observation prediction. Additionally, we demonstrate the robustness of our relational model in continually changing settings, showcasing its adaptability to distractors and environment size. Our results indicate that the GNN-based latent model outperforms MLP-based baselines, achieving superior performance in sparse-reward tasks and demonstrating better generalization to environmental variations.

## 2 BACKGROUND

In this section, we provide the necessary background to understand our approach. We briefly recap the fundamentals of RL, MDPs, and POMDPs, then delve deeper into state abstractions. Subsequently, we formally introduce self-predictive and Observation-Predictive (OP) abstractions, which we use to build our method.

### 2.1 MDPs, POMDPs AND REINFORCEMENT LEARNING

A discounted MDP (Puterman, 2014) is represented by a tuple $\mathcal{M} = (\mathcal{S}, \mathcal{A}, P, R, \gamma, \mu)$. At each time step $t$, an agent observes the state $s_t \sim \mathcal{S}$ of the environment and chooses an action $a_t \sim \mathcal{A}$ using a policy $\pi(a_t \mid s_t)$ to transition into a new state $s_{t+1}$. The dynamics govern the transitions function $P : \mathcal{S} \times \mathcal{A} \times \mathcal{S} \to [0, 1]$, and for each transition, the agent receives a reward according to the reward function $R : \mathcal{S} \times \mathcal{A} \to \mathbb{R}$. The agent's objective is to maximize the expected cumulative discounted reward over an infinite horizon:

$$\max_{\pi} \mathbb{E}_{s_{t+1}\sim P(.|s_t,a_t),a_t\sim\pi(.|s_t)}\left[\sum_{t=0}^{\infty}\gamma^{t-1}r_t\right] \tag{1}$$

where $\gamma \in [0, 1]$ is the discount factor, and the starting state $s_0$ is sampled from the initial state distribution distribution $s_0 \sim \mu(s_0)$.

**Value-based methods** learn an optimal state-action value function $Q^*(s, a)$, the expected return after starting in state $s$ and taking action $a$, by repeatedly performing two steps till convergence: (i) **Policy Evaluation:** computing a value function $Q^\pi(s, a)$ quantifying the expected return after taking action $a$ in state $s$: $Q^\pi(s, a) = \mathbb{E}_\pi\big[\sum_{i=t}^{\infty}\gamma^{i-t}r_{i+1} \mid s_t = s, a_t = a\big]$; and (ii) **Policy Improvement:** learning a new value function from which actions can be greedily selected to maximize $Q^\pi(s, a)$: $\pi'(s_t) \in \arg\max_{a_t\in\mathcal{A}} Q^\pi(s_t, a_t)$

In many real-world scenarios, the agent cannot fully observe the environment. Such problems are modeled by POMDPs, defined as a tuple $\mathcal{M}_\mathcal{O} = (\mathcal{S}, \mathcal{O}, \mathcal{A}, P, R, \gamma, \mu)$, where the agent has access to observations $o \in \mathcal{O}$ based on the state $s \in \mathcal{S}$. It can utilize a history $h_t := \{o_1, a_1, o_2, a_2, \ldots o_t\} \in \mathcal{H}$, by concatenating observations and actions, where $\mathcal{H}$ represents the set of all possible histories.

Since the agent lacks full observability, maintaining a belief state — a probability distribution over possible states given the history — is essential for optimal decision-making (Kaelbling et al., 1998). However, computing and updating such beliefs for high dimensional environments can quickly become intractable (Subramanian et al., 2022). Therefore, the agent requires a history encoder that maps the history to a Markovian representation $\phi_O : \mathcal{H}_t \to \mathcal{Z}$.

## 2.2 State abstractions, Self-Prediction and Observation-prediction

A Q-function itself can be decomposed into two parts: (i) An encoder that $\phi_{Q^*} : \mathcal{S} \to \mathcal{Z}$, that maps the states to abstract states $z \in \mathcal{Z}$, also known as state abstractions (Li et al., 2006), or latent states (Gelada et al., 2019). (ii) A critic $C : \mathcal{Z} \to \mathcal{Q}$ that predicts the $Q-$ values using the latent state $\mathcal{Z}$. This decomposition requires the latent state-space $\mathcal{Z}$ to have sufficient information to accurately predict $Q^*$, i.e. if $\phi(s_i) = \phi(s_j)$, then it must hold that $Q^*(s_i) = Q^*(s_j)$. We can additionally incentivize the latent state to predict the one-step transition probabilities(Equation (ZP)) and rewards (Equation (RP)), thereby preserving the environment's dynamics in the latent space. Equation (ZP) ensures that the latent state is predictive of the subsequent latent state by mapping the joint latent state-action space to a distribution over the latent space $\Delta(\mathcal{Z})$. Consequently, such abstractions are **self-predictive abstractions**, learned using a latent model trained to predict the next latent state (Grill et al., 2020; Guo et al., 2020).

$$\exists P_z : \mathcal{Z} \times \mathcal{A} \to \Delta(\mathcal{Z}) \;\; s.t. \;\; P(z_{t+1} \mid s_t, a_t) = P_z(z_{t+1} \mid \phi_L(s_t), a_t) \tag{ZP}$$

$$\exists P_z : \mathcal{Z} \times \mathcal{A} \to \mathbb{R} \;\; s.t. \;\; \mathbb{E}(r_{t+1} \mid h_t, a_t) = R_z(\phi_L(h_t, a_t)) \tag{RP}$$

For POMDPs, we can extend the state encoder to belief encoder $\phi_O$ to produce a *history abstraction* $z = \phi_O(h) \in \mathcal{Z}$. This encoder satisfies as additional recurrent condition to ensure belief reconstruction:

$$\exists \psi_z : \mathcal{Z} \times \mathcal{A} \times \mathcal{O} \to \mathcal{Z} \;\; s.t. \;\; \phi(h_{t+1}) = \psi_z(\phi_O(h_t), a_t, o_{t+1}) \tag{Rec}$$

Furthermore, such abstractions should additionally satisfy a variant of Equation (ZP), called *Observation-prediction*, ensuring that the latent state along with the action is sufficient to predict the distribution over the subsequent observations (Equation (OP)):

$$\exists P_o : \mathcal{Z} \times \mathcal{A} \to \Delta(\mathcal{O}) \;\; s.t. \;\; P(o_{t+1} \mid h_t, a_t) = P_o(o_{t+1} \mid \phi_O(h_t), a_t) \tag{OP}$$

## 3 METHOD

In this section, we motivate and outline our method. We present the general idea of incorporating additional structure across batches of observations and the inter-trajectory transfer it enables. We then argue how capturing structure across batches is particularly beneficial for tasks with subtask decompositions, especially in a Sparse Reward environment. We then outline our architecture that incentivizes the belief encoder to produce such histories.

### 3.1 RELATIONAL TASK DECOMPOSITION

Complex RL tasks often involve multiple subtasks. In sparse-reward MDPs, these subtasks are crucial but unrewarded steps, making learning challenging due to the delayed feedback. A vital requirement for credit assignment is to model the relationships across these subtasks to assign credit to the crucial state-action pairs. A state abstraction that preserves the optimal Q-value must enable the agent to disentangle latent states corresponding to these crucial ground states.

The intuition behind our approach is that trajectories corresponding to a single subtask exhibit correlations. In addition to the global one-step transition dynamics captured by self- and observation-predictive objectives, local structure among subtasks can be leveraged in the latent space. For example, consider the MDP shown in Section 3.1 where the agent must follow a goal-directed reward to the goal-state $S_5$. The reward includes a small cost per step to the agent and a large reward for reaching the goal. Therefore, the agent must discover the shortest path to reach the goal.

We highlight two example trajectories $\tau_1 = \{S_1, R, S_3, U, S_5\}$ (illustrated in purple) and $\tau_2 = \{S_2, R, S_4, U, S_5\}$ (shown in red). On reaching the goal $S_5$, it gets a reward of $1 - k \times \text{n\_steps}$, where $k \in [0, 1)$. It incentivizes the agent to reach the shortest path to the goal. The two trajectories involve two steps to the goal and accumulate the same return since they both comprise 2-steps to the goal. Although trained solely on data from $\tau_1$, a predictive model capable of capturing relational similarities between these trajectories can generalize to $\tau_2$ by capturing local similarities between these trajectories. For instance, the relationship between $S_3$ and $S_5$ in $\tau_1$ parallels the relationship between $S_4$ and $S_5$ in $\tau_2$.

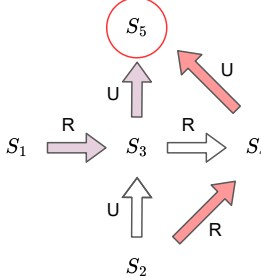

Figure 2: **Example MDP.** The agent must navigate to the goal $S_5$ by maximizing a goal-conditioned reward and minimizing the cost per step. At the start of the episode, the agent can spawn in any of the other states $\{S_1, S_2, S_3, S_4\}$. From each state, it can either go right $R$ or up $U$.

Let us extend this to the POMDP setting, where the agent does not directly observe the states. Instead, it receives partial observations corresponding to these states. The trajectories in this POMDP now correspond to histories of observations, actions, and rewards $h_1 = \{o_1, a_1, r_1, \ldots o_5\}$ and $h_2 = \{o_2, a_2, r_2, \ldots o_5\}$. Here, the observations $o_1, o_2, \ldots$ are partial representations of the states $S_1, S_2, \ldots$, and the goal is to navigate towards the final observation corresponding to $S_5$. Since the agent only observes part of the state, it must infer relationships and similarities between different observation sequences. As in the MDP case, the agent benefits from recognizing relational similarities between these histories to generalize across subtasks.

**Proposition 3.1.** *Let $h_1, \ldots, h_n \in \mathcal{H}$ be histories from similar subtasks in a POMDP, with corresponding next observations $o'_1, \ldots, o'_n \in \mathcal{O}$. Let $\phi : \mathcal{H} \to \mathcal{Z}$ be a Lipschitz continuous function with constant $L_\phi > 0$, mapping histories to embeddings $z_i = \phi(h_i)$. Let $f : \mathcal{Z}^n \to \mathcal{O}$ be a Lipschitz continuous model with constant $L_f > 0$, predicting $o'_{pred} = f(z_1, \ldots, z_n)$.*

*Assume the histories $h_i$ are similar, i.e., $d_{\mathcal{H}}(h_i, h_j) \leq \delta$ for all $i, j$, where $d_{\mathcal{H}}$ measures the distance between histories.*

*Then, minimizing the squared error loss*

$$\mathcal{L} = \|o'_{pred} - o'_i\|^2,$$

*for any $i$, ensures that the prediction error is bounded:*

$$\mathcal{L} \leq (L_f L_\phi n \delta + \epsilon_i)^2,$$

*where $\epsilon_i$ accounts for model approximation errors or inherent noise.*

We sketch this proposition more intuitively by considering the trajectories in Figure 2 as histories. Since transitions from state $S_3 \rightarrow S_5$ and $S_4 \rightarrow S_5$ share a similar relational structure, the embeddings $z_1 = \phi(\{S_3, U, S_5\})$ and $z_2 = \phi(\{S_4, U, S_5\})$ will be close in the latent space. Training a model to minimize the loss $\mathcal{L}$ by reasoning over both these trajectories ensures that the model generalizes between these subtasks, capturing the similarities between these histories. We do this using a GNN. Please refer to Appendix A.1 for a more detailed proof sketch.

### 3.2 Observation Prediction using a Graph-based Latent Model

Our method comprises three key components, illustrated in Figure 3:

1. **Encoder** ($\phi$) that maps histories to latent representations $z$.

2. **Model** ($\psi$) that captures relationships among history embeddings.

3. **RL network** ($\pi_\theta$ or $q_\theta$) that uses $z$ for either learning a policy, or a $Q$- function depending on the method that we utilize.

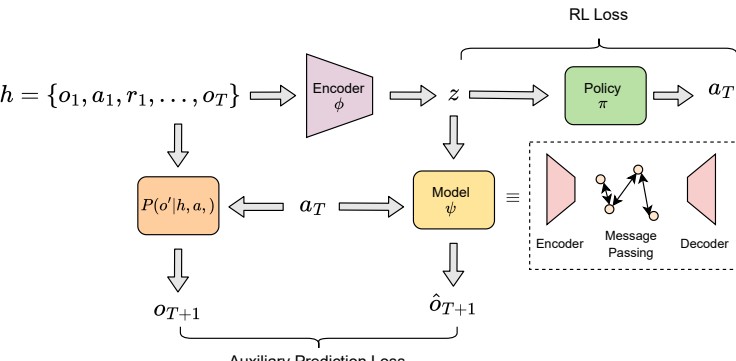

Figure 3: **Training Setup.** The LSTM generates embeddings using observation history, actions, and rewards, capturing temporal dependencies to create a belief state $z$. The policy network uses this to select the next action. For a value-based agent in a discrete action space, this would be a critic network that outputs values over discrete actions. Then, the algorithm greedily selects the action with the highest value. During optimization, the structured model - A GNN- reasons over a batch of latent states and corresponding actions to predict the subsequent observations. This is compared against the corresponding next observations to create the auxiliary prediction loss.

**Encoder and Policy Network.** The encoder $\phi$ maps the history of observations to a latent state $z = \phi(h_t)$. In a POMDP, this is either a recurrent encoder (Subramanian et al., 2022) or possibly a transformer with a sufficiently large context window (Esslinger et al., 2022). Any RL agent can now use this latent state. In both these cases, the policy $\pi(a_t | z_t)$ takes the latent state $z_t$ as an input and outputs an action $a_t$. Value-based methods use a critic network that outputs values for each action for a given $z$ and greedily selects the action with the maximum value.

**Graph construction.** The latent model $\psi$ is a self-predictive model to enhance representation learning. To capture relational structure within the latent space, we consider a batch of latent states $Z = [z_1, \ldots, z_T]$ and corresponding actions $\{a_1, \ldots a_T\}$. We convert these actions to one-hot vectors and then concatenate them to form node features $\{(z_1, a_1), \ldots, (z_T, a_T)\}$. Then, we construct a $m$-nearest neighbors graph on these with $m = 4$ using the Euclidean distance between the node features.

**Message Passing.** After constructing the graph, the nodes with actions as attributes are passed through two message-passing layers. During this phase, each node in the graph updates its state by aggregating information from its neighboring nodes. Firstly, for each node, the features of its neighboring nodes are aggregated by concatenating the features of the source node $x_i$ and the target node $x_j$. This concatenated vector is then passed through an MLP consisting of two fully connected layers with a ReLU activation function in between, transforming the combined features to capture more complex interactions. The result of this MLP is then used to update the target node's features.

**Observation-Prediction and training.** After the message-passing steps, the updated node features are decoded to produce the final node representations. The output of the network has the same dimensionality as the flattened observation dimensions, and therefore, allows the graph to predict a batch of the subsequent observations $\{\hat{o}_2, \ldots, \hat{o}_{t+1}\}$ by reasoning across the batch of $T$ observations and actions. The output of the GNN is then compared with the corresponding ground-truth observations $\{o_2, \ldots, o_{t+1}\}$ present in the buffer during training to create an auxiliary loss. This loss is jointly optimized along with the RL loss from the policy or critic network. As a result, we can train the encoder ($\phi$), the model ($\psi$), and the policy ($\pi$) together during the optimization procedure.

$$\{\hat{o}_2, \ldots, \hat{o}_{T+1}\} = \psi(\{[z_j, a_j]\}_{j=1}^T)$$

This output is trained using the Mean-Squared Error (MSE) loss between the predicted outputs $\{\hat{o}'_1, \ldots, \hat{o}_T\}$ and the actual next observation $\{\hat{o}_1, \ldots, \hat{o}_T\}$ sampled from the batch forming the representation learning auxiliary loss:

$$\mathcal{L}_{\text{OP}} = \sum_{t=1}^{T} \|\hat{o}_{t+1} - o_{t+1}\|^2$$

In principle, this objective is agnostic to the RL objective and, therefore, can be combined with any RL algorithm. We demonstrate an example of using our model with a policy-gradient algorithm in Algorithm 1.

**Reward Module.** For environments with multiple subtasks and sparse rewards, OP alone is insufficient (Ni et al., 2024). Instead, it must be combined with an explicit reward prediction using the latent state and action. For these environments, we utilize a two-layer MLP for such a module in addition to the latent model and train it using a phased training procedure, where the reward module is optimized separately from the end-to-end optimization of the bellman and representation learning loss. Instead, we interleave the optimization of the reward prediction from the representation learning modules by optimizing them one after the other.

## 4 EXPERIMENTS

In this section, we empirically investigate the effectiveness of our structured latent model. We employ the Minigrid suite (Chevalier-Boisvert et al., 2023), which consists of a series of mini-levels designed to test various aspects of learning and adaptation. The RL agent in our experiments is the R2D2 agent (Kapturowski et al., 2019), including a recurrent replay buffer with uniform sampling. Our hyperparameters can be found in A.2. In the following paragraphs, we divide our analysis based on specific research questions. Our presented results have been performed across 5 seeds with the aggregated IQMs (Agarwal et al., 2021).

---

**Algorithm 1** Training Procedure with a value-based agent

---

**Require:** Initialized encoder $\phi$, policy network $\pi$, auxiliary graph model $\psi$
1: **while** not converged **do**
2:     **Collect Trajectories** using policy $\pi(a_t \mid z_t)$:
3:       Collect experiences $\tau = \{(o_t, a_t, r_t, o_{t+1})\}$
4:       Compute $z_t = \phi(o_t)$, $z_{t+1} = \phi(s_{t+1})$
5:       Collect experiences $\tau = \{(o_t, a_t, r_t, o_{t+1})\}$
6:       Compute $z_t = \phi(o_t)$, $z_{t+1} = \phi(o_{t+1})$
7:     **Compute RL Loss**:
8:       Compute target values: $V_{target} = r_t + \gamma V(z_{t+1})$
9:       Estimate Q-values: $Q(z_t, a_t) \leftarrow Q(z_t, a_t)$
10:     $\mathcal{L}_{\text{RL}} = \frac{1}{N} \sum_t \left(Q(z_t, a_t) - V_{target}\right)^2$
11:     **Compute Observation-Prediction Loss**:
12:       Construct graphs $G_t$ from $z_t$
13:       Predict $\hat{o}_{t+1} = \psi(G_t, a_t)$
14:     $\mathcal{L}_{\text{OP}} = \sum_t \|\hat{o}_{t+1} - o_{t+1}\|^2$
15:     **Update Parameters**:
16:     $\mathcal{L} = \mathcal{L}_{\text{RL}} + \lambda \mathcal{L}_{\text{OP}}$
17:     Minimize $\mathcal{L}$ w.r.t. $\phi$, $\pi$, $\psi$
18: **end while**

---

**Performance on static environments.** We first evaluate our model (`Graph_OP`) on selected environments in Minigrid. Our baselines are the observation predictive algorithm (`min-OP`) and the observation and reward prediction algorithm (`min-AIS`) (Ni et al., 2024). `min-OP` follows the same pipeline but uses an MLP for the observation prediction task. The MLP predicts the subsequent observation for each latent state in a batch and does not use relational reasoning for the whole batch. `min-AIS`, on the other hand, extends `min-OP` by predicting the subsequent reward in addition to the observation, improving performance in environments where observation prediction alone is insufficient for effective representation learning. The critical distinction between our method and these baselines is how they process the latent observations and associated actions. In the MLP-based baselines, each combination of latent observation and action is processed independently to predict the subsequent observation. By contrast, our GNN-based approach first constructs a graph over all the latent observation-action pairs in the batch, applies message passing across the graph to model relational dependencies, and then predicts the subsequent observations for each element. Therefore, the performance difference between the baselines and our method primarily comes from this privileged reasoning. We consider environments with subtasks from the Minigrid suite challenging without representation learning and particularly challenging for observation prediction. Please note that R2D2, without representation learning, fails to accumulate notable returns in these environments, as indicated by the curves in Ni et al. (2024). Moreover, we run each environment until the baselines demonstrate convergent behavior. Based on the learning curves provided by Ni et al. (2024), we narrow down the environments to the following four static ones:

1. `MiniGrid-DoorKey-8x8-v0`: The agent must pick up a key to unlock a door and reach the green goal in a $8 \times 8$ grid.

2. `MiniGrid-ObstructedMaze-1Dl-v0`: A blue ball is hidden in a maze with two rooms. A locked door separates the two rooms, and a ball obstructs the doors. The keys are hidden in boxes.

3. `MiniGrid-KeyCorridorS3R2-v0`: The agent has to pick up an object behind a locked door. The key is hidden in another room, and the agent has to explore the environment to find it.

4. `MiniGrid-UnlockPickup-v0`: The agent must pick up a box behind a locked door in another room.

These environments share the commonality of subtasks the agent needs to solve before reaching the goal. Apart from the DoorKey environment, all others require additional reward

prediction due to the sparsity of the reward in the original task. Consequently, we incorporate an additional reward-prediction module with our graph prediction (`Graph_AIS`).

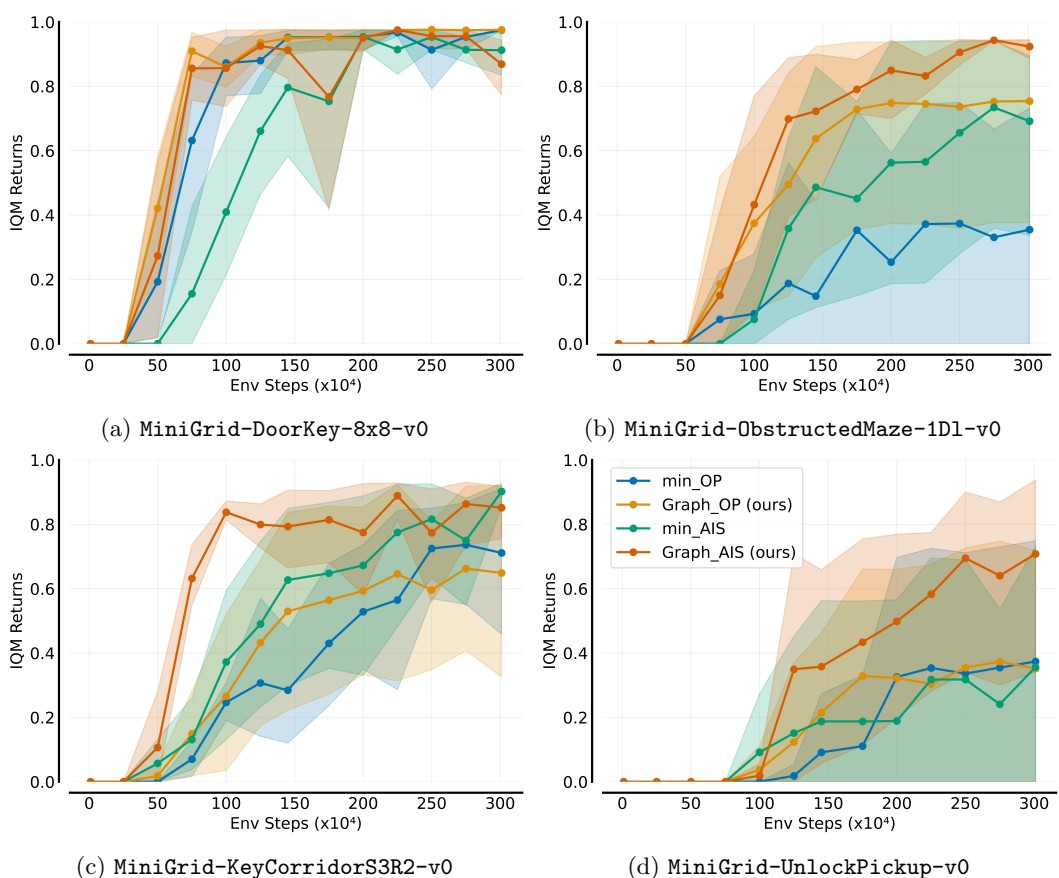

(a) `MiniGrid-DoorKey-8x8-v0`

(b) `MiniGrid-ObstructedMaze-1Dl-v0`

(c) `MiniGrid-KeyCorridorS3R2-v0`

(d) `MiniGrid-UnlockPickup-v0`

Figure 4: IQM and quartiles of Performances on static environments.

Our results are presented in Figure 4. Overall, the `Graph`-based representation learning methods outperform the MLP-based techniques in most cases. For environments where observation prediction struggles with long-term dependencies, the combination of Graph-based observation prediction and reward prediction – `Graph_AIS` – consistently outperforms the baselines. This reiterates the inefficiencies of pure observation prediction in such environments since the reward is highly sparse in these subtasks.

**Adapting to environment changes.** A crucial outcome of Proposition 3.1 would be the ability of our method to extrapolate the learned prediction across environmental changes insofar as these changes share some similarity with data seen already. We investigate this by creating a scenario where an agent must continually adapt to environmental variations. We introduce changes to `MiniGrid-DoorKey-8x8-v0` by changing: (i) **Number of keys:** We introduce distractions in the form of additional colorless keys, forcing the agent to focus on the colored key. The number of distractors remains constant for each episode, but their location changes after the reset. (ii) **Size:** We periodically increase the size of the environment to investigate how well the agent adapts to the increase in the number of states.

Figure 5 shows the performance of `Graph_OP` against `min_OP` for different types of changes. Figure 5(a) demonstrates the agent's performance when distractors are added after $800K$ steps, and Figure 5(b) shows the adaptation to increase in size after $1M$ steps. We introduce additional dimensions of hardness by combining these changes. Figure 5(c) shows the scenario in which the grid increases in size every $1M$ step, and a distractor is simultaneously added. Finally, Figure 5(d) shows the scenario in which the agents must adapt to a new distractor every $600K$ step and a size increment every $1M$ step in the bottom right figure.

As expected, both methods' performance generally degrades when changes occur, and recovery from these changes becomes increasingly difficult as we increase hardness. As a result, in Figure 5(d), neither method has enough time to return to stable performance. In most of these scenarios, `Graph_OP` remains consistently more robust performance and outperforms `min_OP`. The impact of distractions seems more pronounced than size, as shown in Figure 5(a).

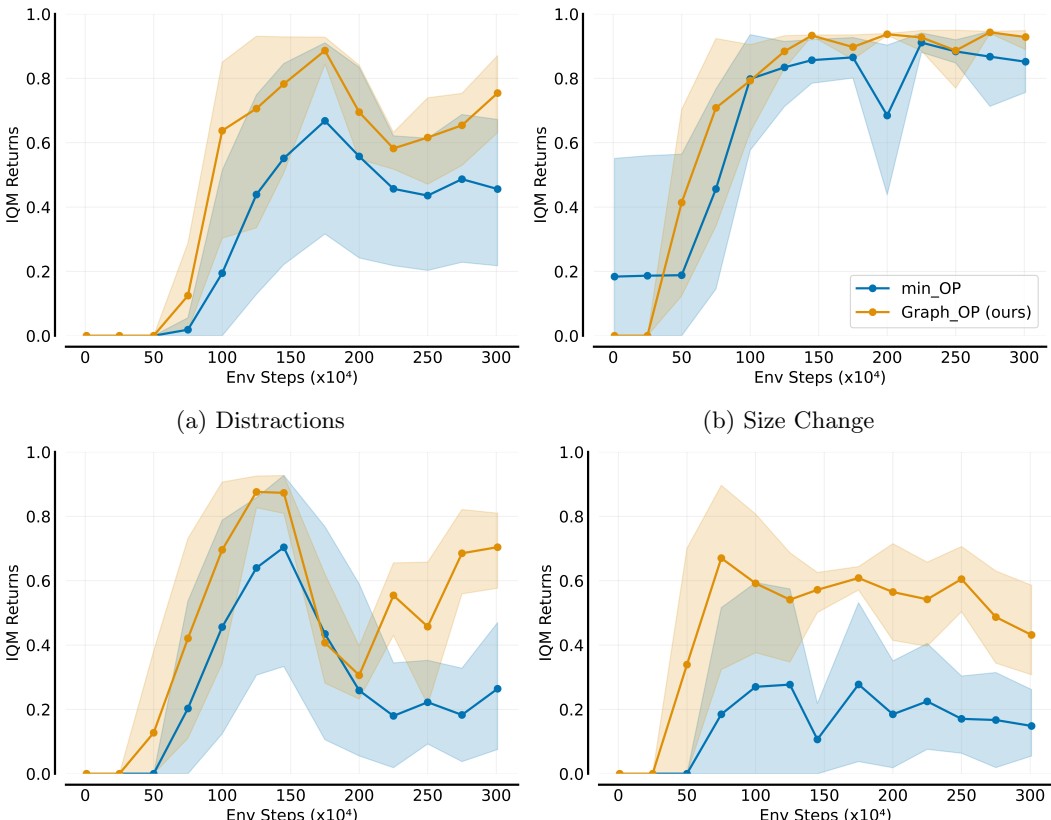

(a) Distractions

(b) Size Change

(c) Simultaneous change in size and distractions    (d) Interleaved change in size and distractions

Figure 5: Performances on Dynamic Variations of `MiniGrid-DoorKey-8x8-v0`.

Compound changes particularly impact both methods since the size change forces the agent to explore more, while the distractors force the agent to focus on the right kind of key. Given that in DoorKey, the agent has to traverse a sub-goal of getting to a key before reaching a door and then going to the goal, changing the size and adding distractors together degrades performance faster. In both cases, the graph-based agent `Graph_OP` is more robust to the changes than the MLP baseline. This highlights the particular advantage relational inductive bias offers: it allows the state representations to model relationships between trajectories and the one-step temporal consistency of self-prediction.

## 5    RELATED WORK

Our work touches upon three crucial areas in RL: *Abstractions, GNNs in RL, and incorporating structure in RL.* summarized below.

**State and History Abstractions in RL.**    State abstractions constitute an active area in RL, and a complete categorization of approaches is beyond the scope of this work. Model-irrelevance has been studied under a variety of techniques, such as bi-simulation (Ferns et al., 2004; Gelada et al., 2019; Castro et al., 2021; Hansen-Estruch et al., 2022; Lan &

Agarwal, 2023), variational inference (Eysenbach et al., 2021; Ghugare et al., 2023), and successor features (Dayan, 1993; Barreto et al., 2017; Borsa et al., 2019; Lehnert & Littman, 2020; Scarpellini et al., 2024). Self-predictive representations have been a separate line of work (Guo et al., 2020; Grill et al., 2020; Schrittwieser et al., 2020; Schwarzer et al., 2021; Hansen et al., 2022; Ghugare et al., 2023; Zhao et al., 2023) with increasing interest in understanding how these objectives behave (Tang et al., 2023; Ni et al., 2024; Fang & Stachenfeld, 2024; Voelcker et al., 2024; Khetarpal et al., 2024). Observation predictive representations have been used to formulate belief states (Kaelbling et al., 1998; Wayne et al., 2018; Hafner et al., 2019; Han et al., 2020; Lee et al., 2020) and predictive state representations (Littman et al., 2001; Zhang et al., 2019), and are also related to observation reconstruction objectives commonly used for improving sample efficiency Yarats et al. (2021). Our work adds to this line of work by exploring how the self-predictive objective can capture relational structure in the latent space.

**Structure in RL.** Structural decompositions can be useful as inductive biases for various purposes (Mohan et al., 2024). Our work assumes a relational decomposition in joint state-action space. Such assumptions have previously been applied through modeling frameworks such as Relational MDPs (Dzeroski et al., 2001; Guestrin et al., 2003) and object-oriented MDPs (Diuk et al., 2008). However, we neither model entities in the environment separately nor handcraft any form of first-order representation in the value function (Guestrin et al., 2003; Fern et al., 2006; Joshi & Khardon, 2011). Instead, we reason across trajectories using a GNN to model relationships.

**GNNs in RL.** GNNs have increasingly been used in RL, such as modeling environments (Chen et al., 2020; Chadalapaka et al., 2023), agent's morphology in embodied control (Wang et al., 2018; Oliva et al., 2022), relationships between different action sets (Jain et al., 2021), and concurrent policy optimization method (Wang & van Hoof, 2022). We share similarities to methods that use GNNs as structured models, used for applications such as learning the latent transition dynamics in simple manipulation tasks (Kipf et al., 2020), the dynamics of joints of physical bodies (Sanchez-Gonzalez et al., 2020), obtaining object-centric representations from images and RRT planners (Driess et al., 2022), or computing intrinsic reward and online planning (Sancaktar et al., 2022). We add to this line of work by using GNNs for observation-prediction. Although Transformers have also been used for learning state representations (Zhu et al., 2022) and state-action representations (Zheng et al., 2024), they require substantial data and computational resources, often making them less practical in data-scarce RL settings. In contrast, GNNs effectively leverage structural properties in relational tasks, providing an efficient alternative for relational reasoning in reinforcement learning.

## 6 Conclusion and Future Work

Using a structured latent model to investigate the impact of relational inductive biases, Using a structured latent model to investigate the impact of relational inductive biases, we incorporated a GNN to capture the similarity between the latent space belief representations produced by a recurrent encoder. Our experiments on a relevant subset of Minigrid tasks demonstrated that agents utilizing this latent space exhibit improved performance and the learned representations tend to be more robust to changes in size and against added distractions. Although effective, our approach has been evaluated only on discrete action spaces and requires further investigations on continuous action spaces in environments such as robotic control (Freeman et al., 2021; Todorov et al., 2012), and on more complicated navigation topologies such as those found in Cobbe et al. (2020); Samvelyan et al. (2021). Additionally, we want to incorporate more algorithms since the current framework is agnostic to the RL algorithm. Finally, we want to extend our method to 3D point clouds to capture granular structure. Despite these limitations, our current findings offer a foundation for future research, and addressing these challenges will be crucial to advancing the capabilities of graph-based latent models in RL.

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

# A APPENDIX

## A.1 PROOF SKETCH OF PROPOSITION 3.1

In this section, we provide a theoretical foundation for the generalization capability of our proposed method. We formalize the relationship between subtask similarity and the embeddings learned by the GNN-based model. We restate the proposition in detail below:

**Proposition A.1.** *Let $h_1, h_2, \ldots, h_n \in \mathcal{H}$ be histories sampled from individual subtasks at different time steps in a POMDP, and let $o'_1, o'_2, \ldots, o'_n \in \mathcal{O}$ be the corresponding next observations. Let $\phi : \mathcal{H} \to \mathcal{Z}$ be a belief function mapping histories to embeddings $z_i = \phi(h_i)$. Assume that $\phi$ is Lipschitz continuous; that is, there exists a constant $L_\phi > 0$ such that for all $i, j$:*

$$\|z_i - z_j\| \leq L_\phi \cdot d_{\mathcal{H}}(h_i, h_j),$$

*where $d_{\mathcal{H}} : \mathcal{H} \times \mathcal{H} \to \mathbb{R}_{\geq 0}$ is a distance metric on $\mathcal{H}$. Let $f : \mathcal{Z}^n \to \mathcal{O}$ be a model that predicts an observation $o'_{pred} = f(z_1, \ldots, z_n)$. Assume that $f$ is Lipschitz continuous with constant $L_f > 0$.*

*Then,*

$$\left( \max_{i,j} d_{\mathcal{H}}(h_i, h_j) \leq \delta \right) \implies \mathcal{L}(o'_{pred}, o'_i) \leq \left( L_f L_\phi n \delta + \epsilon_i \right)^2,$$

*where $\epsilon_i$ represents the inherent error due to model approximation or noise.*

**Proof Sketch.**

**Step 1: Lipschitz Continuity of $\phi$.** Since $\phi$ is Lipschitz continuous:

$$\|z_i - z_j\| \leq L_\phi \cdot d_{\mathcal{H}}(h_i, h_j) \leq L_\phi \delta \quad \text{for all } i, j.$$

**Step 2: Bounding Differences in Embeddings.** The maximum distance between any pair of embeddings $z_i, z_j$ is bounded:

$$\|z_i - z_j\| \leq L_\phi \delta.$$

**Step 3: Lipschitz Continuity of $f$.** Applying $f$ to embeddings $z_1, \ldots, z_n$ and another set $z'_1, \ldots, z'_n$ (which in this case are $z_j$, since embeddings are close):

$$\|f(z_1, \ldots, z_n) - f(z'_1, \ldots, z'_n)\| \leq L_f \sum_{k=1}^{n} \|z_k - z_j\|.$$

Since $\|z_k - z_j\| \leq L_\phi \delta$:

$$\|f(z_1, \ldots, z_n) - f(z'_1, \ldots, z'_n)\| \leq L_f L_\phi n \delta.$$

**Step 4: Relating to the True Observation.** Assuming $o'_j = f(z_j, \ldots, z_j) + \epsilon_j$, where $\epsilon_j$ accounts for model approximation error or noise. Then, for any $i$:

$$\|o'_{\text{pred}} - o'_i\| \leq \|o'_{\text{pred}} - o'_j\| + \|o'_j - o'_i\|.$$

Since $o'_{\text{pred}}$ is close to $o'_j$ due to the bound from Step 3, and $o'_j$ is close to $o'_i$ if $o'_i \approx o'_j$.

*Justification:* The model $f$ processes a batch of embeddings $z_1, \ldots, z_n$ to predict the next observation $o'_{\text{pred}}$. When we input identical embeddings $z_j$ into $f$, i.e., $f(z_j, \ldots, z_j)$, the model effectively focuses on the information contained in $z_j$ without interference from variations in other embeddings. Given that $z_j$ represents the embedding of history $h_j$, it is reasonable to expect that $f(z_j, \ldots, z_j)$ approximates the true next observation $o'_j$, up to some approximation error $\epsilon_j$.

**Step 5: Bounding the Prediction Error.** Combining the above:

$$\|o'_{\text{pred}} - o'_i\| \leq L_f L_\phi n \delta + \epsilon_i,$$

where $\epsilon_i$ accounts for discrepancies between $o'_i$ and $o'_j$ and any inherent noise.

**Step 6: Squared Error Loss.** Therefore:

$$\mathcal{L}(o'_{\text{pred}}, o'_i) = \|o'_{\text{pred}} - o'_i\|^2 \leq (L_f L_\phi n \delta + \epsilon_i)^2.$$

Hence, minimizing the squared error loss under the Lipschitz continuity of $\phi$ and $f$ under the assumption of similar histories ensures that small differences in histories lead to proportionally small prediction errors. This confirms that our method effectively leverages relational structures among histories to generalize across subtasks, validating the proposition. $\square$

While the proof establishes an upper bound on the prediction error based on the Lipschitz continuity of $\phi$ and $f$, it's important to consider how minimizing the squared error loss

$$\mathcal{L}(o'_{\text{pred}}, o'_i) = \|o'_{\text{pred}} - o'_i\|^2$$

during training impacts the approximation errors $\epsilon_i$ and the bound.

Minimizing $\mathcal{L}$ reduces the approximation errors $\epsilon_i$, leading to a tighter bound on the prediction error:

$$\mathcal{L}(o'_{\text{pred}}, o'_i) \leq (L_f L_\phi n \delta + \epsilon_i)^2.$$

As $\epsilon_i$ decreases, the bound becomes tighter, enhancing the model's predictive accuracy. This process improves the model's ability to generalize across similar histories and subtasks by effectively capturing relational structures in the data. Therefore, minimizing the loss during training is crucial for achieving the theoretical benefits outlined in the proof.

### A.2 HYPERPARAMETERS AND EXPERIMENTAL DETAILS

| Hyperparameter | Value |
|---|---|
| Discount factor ($\gamma$) | 0.99 |
| Number of environment steps | $3 \times 10^6$ |
| Maximum number of distractors | 4 |
| Maximum size change | $12 \times 12$ |
| Target network update rate ($\tau$) | 0.005 |
| Replay buffer size | $400,000$ |
| Batch size | 256 |
| Learning rate | 0.001 |
| Latent state dimension | 128 |
| Epsilon greedy schedule | exponential$(1.0, 0.05, 400,000)$ |
| R2D2 sequence length | 10 |
| R2D2 burn-in sequence length | 5 |
| $n$-step TD | 5 |
| Training frequency | every 10 environment steps |
| Auxiliary loss coefficient ($\lambda$) | 0.01 |
| Latent state size | 147 |
| Num. neighbors in GNN ($m$) | 4 |
| Num. of message passing steps | 2 |
| Hidden state of Graph model | $147//2 = 73.5$ |

## A.3 Latent Space Trajectories

This section outlines the methodology used to construct visual trajectories in the latent space of the encoder. These visualizations provide insights into how the latent spaces encode task-relevant information across different phases of the agent's trajectory, such as key collection and goal navigation.

To generate these trajectories, we used the checkpoint of a trained encoder and simulated a path to the goal. We then divided this into two phases based on the subtask of key collection:

1. **Phase 1:** trajectory until collection of the key.

2. **Phase 2:** trajectory after collecting the key until the goal.

For each phase, the hidden states produced by the encoder were collected during the execution of the corresponding actions. We then applied Principal Component Analysis (PCA) to reduce the dimensionality of these latent states to three components, enabling visualization in 3D space. The resulting points connect consecutive latent states, forming a trajectory in the latent space. Each connection and corresponding point is color-coded by phase to emphasize transitions between sub-tasks, with the goal state represented as a distinct point in the latent space. This visualization allows a qualitative comparison of how algorithms organize and structure their latent representations for task completion. We now summarize the general observations from these figures.

**Clearer Trajectories in Graph_OP.** The latent trajectories reveal notable differences in how various objectives shape the latent space representations. The Graph_OP method consistently exhibits clearer and smoother trajectories between task phases, such as key collection and goal navigation. This clarity arises from the graph prediction objective, which helps the model learn a well-structured latent space. By focusing on observation prediction, Graph_OP emphasizes encoding the environment's dynamics and transitions between states, resulting in smoother and more structured latent.

**Ruggedness in Graph_AIS.** In contrast, incorporating the reward prediction objective, as seen in Graph_AIS, introduces more ruggedness into the latent trajectories. This ruggedness reflects the aggressive influence of the reward prediction objective, which aligns the latent space with task rewards. While this alignment prioritizes encoding goal-directed information, it often disrupts the smooth structure typically learned by the graph prediction objective. Consequently, the latent trajectories for Graph_AIS are less structured than that of Graph_OP but better aligned with task-relevant rewards.

**Goal State Placement.** Another key observation is the placement of the goal state in the latent space. In Graph_AIS, the goal state appears further away from other latent states compared to Graph_OP. This distinction highlights how the reward prediction objective drives the model to strongly differentiate goal states from other regions of the latent space. This explicit separation facilitates more effective credit assignment, enabling the agent to focus on actions that lead to the goal.

**Why Graph_AIS Outperforms Graph_OP.** Despite the less structured latent space, Graph_AIS generally outperforms Graph_OP. This is because reward alignment ensures that the latent space emphasizes task-relevant features, particularly those associated with long-term planning and goal achievement. Combining the graph and reward prediction objectives enables Graph_AIS to balance relational modeling and goal-directed alignment, improving task performance.

### A.3.1 MiniGrid-DoorKey-8x8-v0

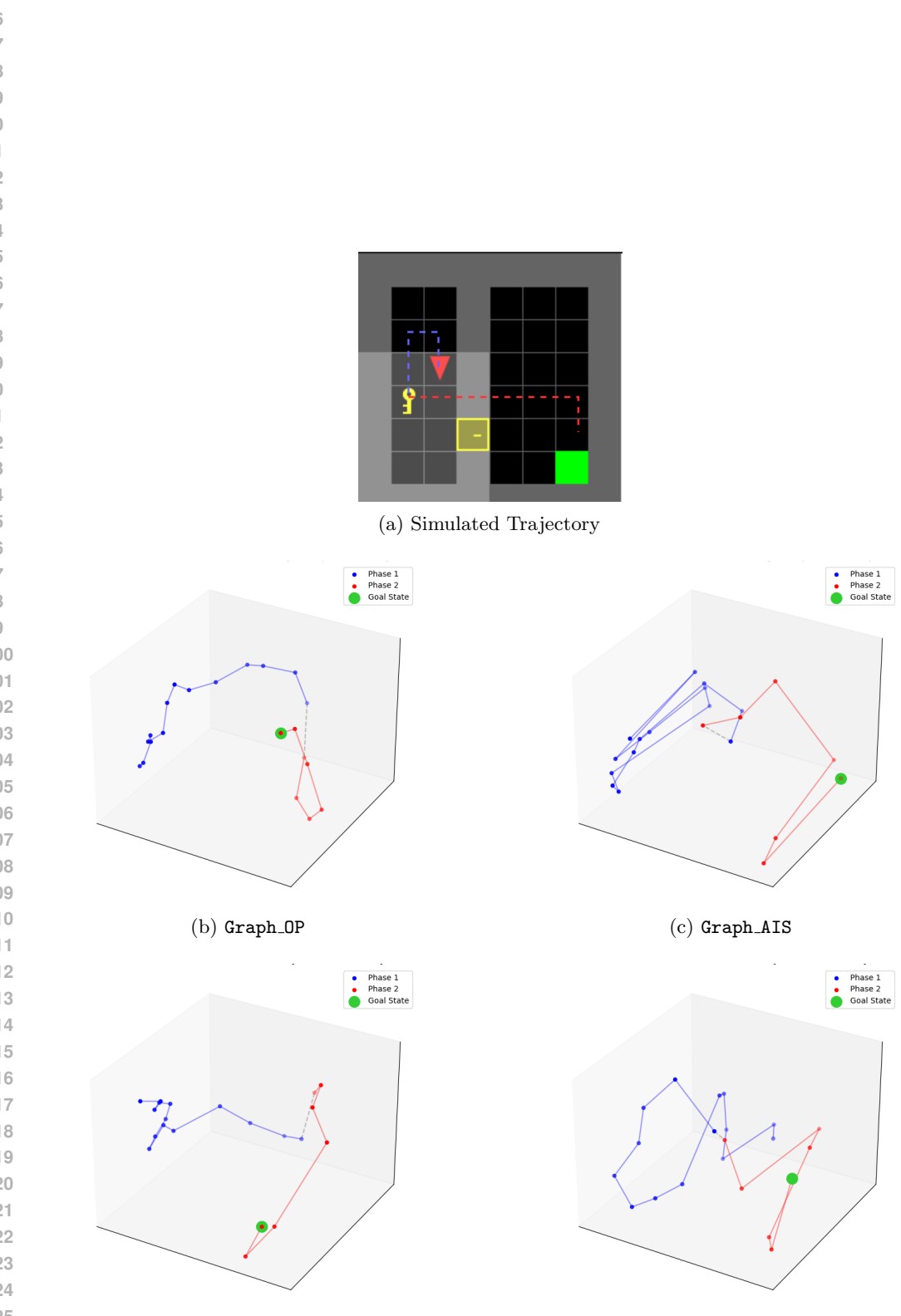

(a) Simulated Trajectory

(b) `Graph_OP`

(c) `Graph_AIS`

(d) `min_OP`

(e) `min_AIS`

### A.3.2 MiniGrid-ObstructedMaze-1Dl-v0

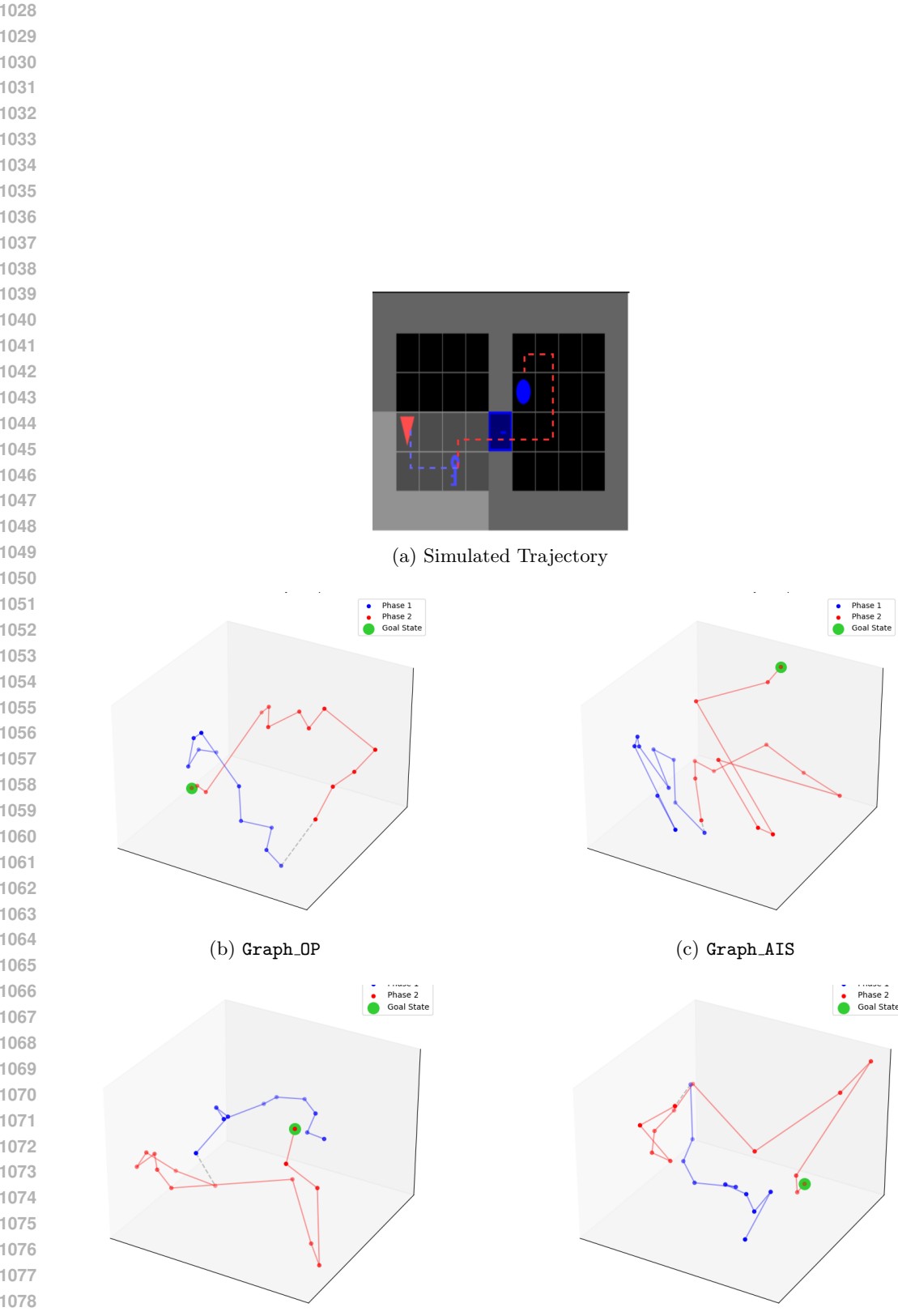

(a) Simulated Trajectory

(b) `Graph_OP`

(c) `Graph_AIS`

(d) `min_OP`

(e) `min_AIS`

### A.3.3  MiniGrid-KeyCorridorS3R2-v0

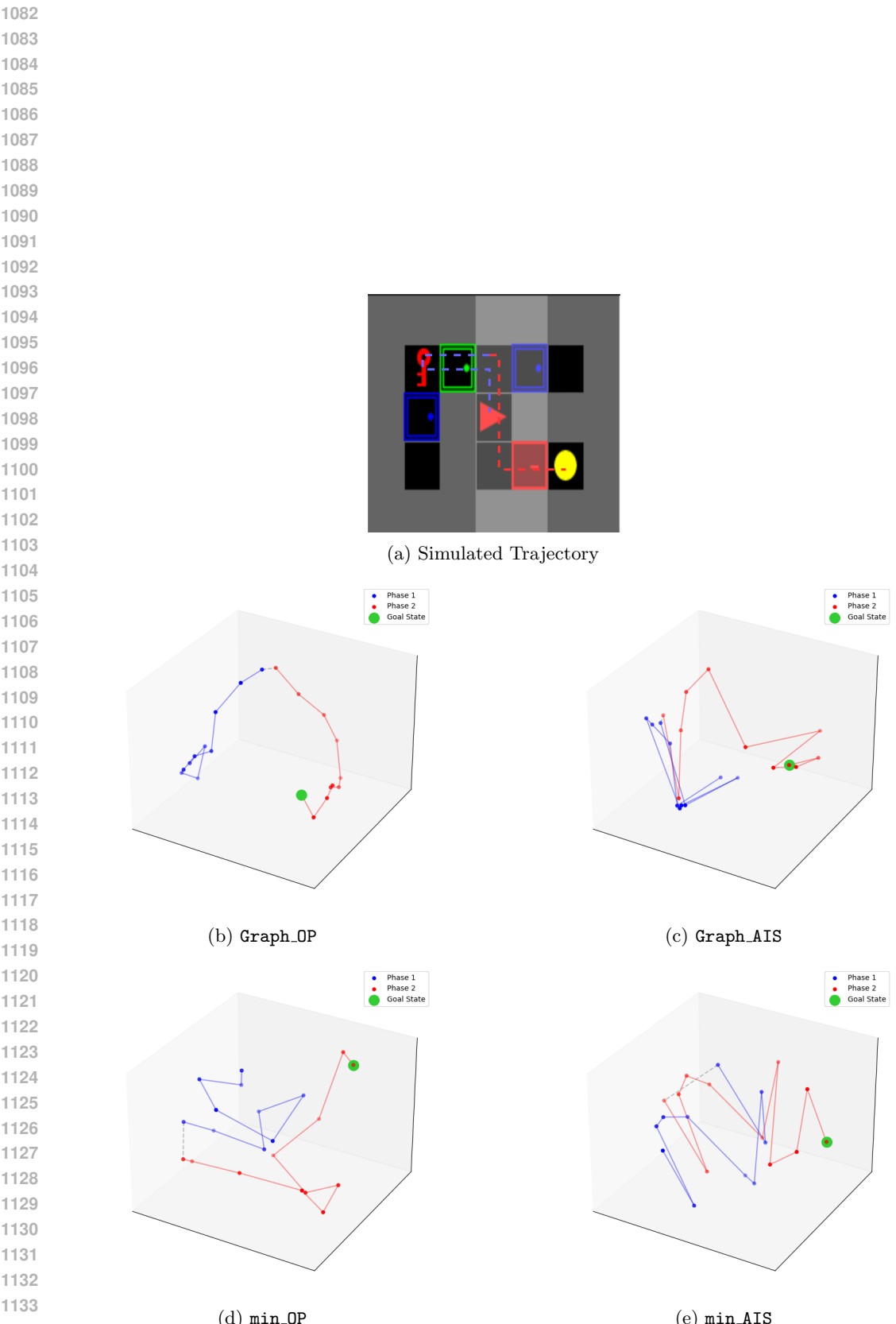

(a) Simulated Trajectory

(b) Graph_OP

(c) Graph_AIS

(d) min_OP

(e) min_AIS

### A.3.4  MiniGrid-UnlockPicup-v0

(a) Simulated Trajectory

(b) `Graph_OP`

(c) `Graph_AIS`

(d) `min_OP`

(e) `min_AIS`

## A.4    Prediction of the Models

In this section, we compare the predictions generated by the MLP-based model and the Graph-based model. To produce the predictions in the figures below, we initialized an agent, loaded the model, critic, and encoder checkpoints, and populated the buffer by interacting with the environment. A minibatch of observations was then sampled from this buffer, and the model was queried to predict the corresponding subsequent observations. The figure compares an observation image from the batch with the predictions from the MLP-based and Graph-based models.

The Graph-based model consistently generates predictions with higher fidelity than the MLP-based model, highlighting the advantages of the GNN's temporal reasoning capabilities. While the MLP model struggles to produce visually accurate reconstructions, it retains vital features such as approximate spatial contrasts and object colors. These features may explain its ability to perform reasonably despite poor visual quality. In contrast, the Graph model produces predictions that closely resemble the original observations, demonstrating its superior ability to leverage temporal relationships across trajectories.

### A.4.1    MiniGrid-DoorKey-8x8-v0

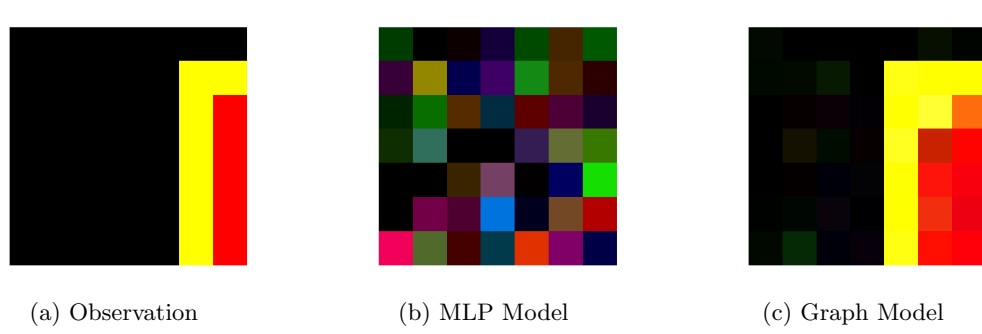

(a) Observation          (b) MLP Model          (c) Graph Model

### A.4.2    MiniGrid-ObstructedMaze-1Dl-v0

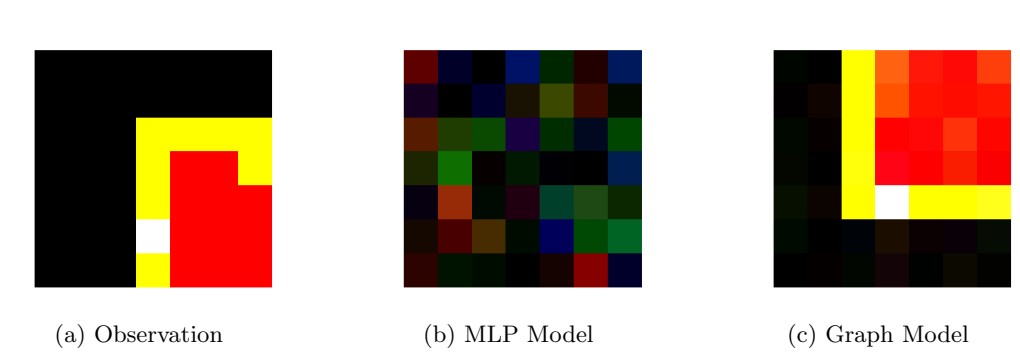

(a) Observation          (b) MLP Model          (c) Graph Model

### A.4.3  MiniGrid-KeyCorridorS3R2-v0

(a) Observation          (b) MLP Model          (c) Graph Model

### A.4.4  MiniGrid-UnlockPickup-v0

(a) Observation          (b) MLP Model          (c) Graph Model

## A.5 DIFFERENT VALUES FOR NEIGHBORS

We ablated the number of neighbors ($m$) used in the graph construction to evaluate its effect on task performance. The results, presented in Appendix A.5, demonstrate that the model is robust to changes in $m$, with similar final returns across $m = 4$, $m = 6$, $m = 8$, and $m = 16$ in most tasks. In the early stages of training, $m = 4$ tends to achieve faster returns, suggesting that smaller graphs may provide more efficient learning initially. However, tasks with more complex relational dependencies, such as `UnlockPickup-v0`, benefit slightly from $m = 6$, indicating that the optimal number of neighbors may be task-specific. Larger values of $m$ introduce more variability in performance for some environments, as evidenced by broader confidence intervals, potentially due to increased noise in the graph. Overall, these results highlight the robustness of the proposed method across different graph configurations, with $m = 4$ serving as a reasonable default choice for most tasks.

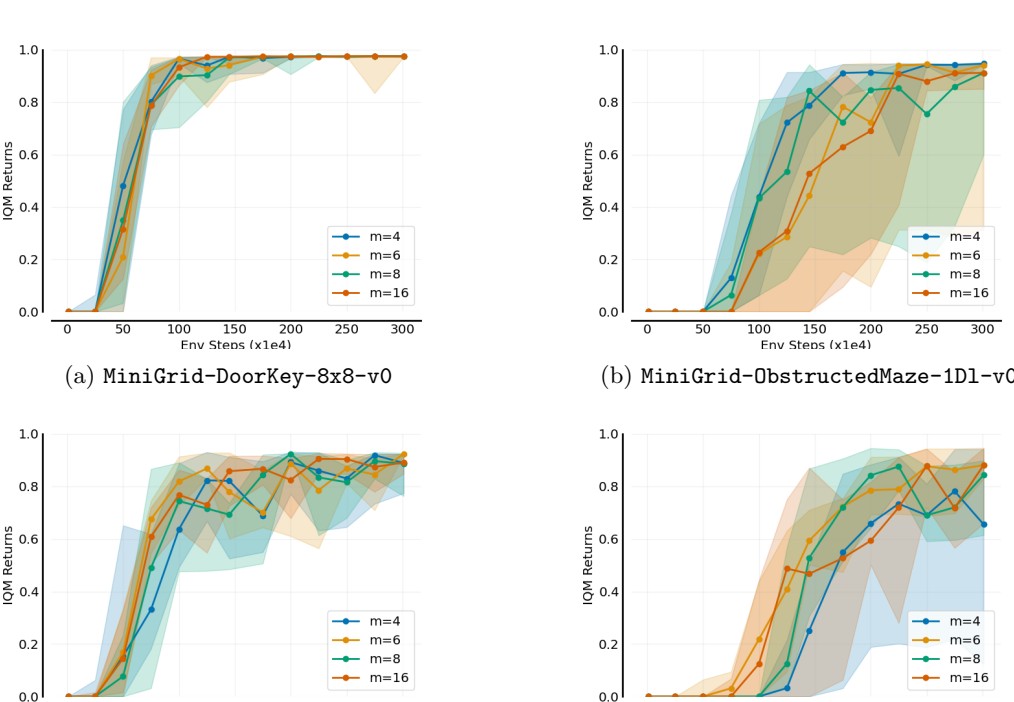

(a) `MiniGrid-DoorKey-8x8-v0`

(b) `MiniGrid-ObstructedMaze-1Dl-v0`

(c) `MiniGrid-KeyCorridorS3R2-v0`

(d) `MiniGrid-UnlockPickup-v0`

## A.6 Isolating the Effect of Relational Reasoning in the GNN

We designed an experiment to isolate this effect and understand whether the GNN's observed benefits arise from its relational reasoning or simply from operating on the entire batch of observations. In our standard setup, the GNN processes a batch of observations by constructing a graph over the entire batch and performing relational reasoning through message passing. By contrast, the baseline MLP independently predicts the next observation for each element in the batch without leveraging relationships across the batch.

We modified the GNN and MLP architectures for this experiment to process mini-batches of 50 observations each sequentially. Specifically, we divided the original batch into 50-unit mini-batches and processed them sequentially. The GNN constructed a graph over each mini-batch and performed relational reasoning with a sparse connection via message passing, while the MLP processed the mini-batches without relational reasoning. After processing each mini-batch, the outputs were concatenated into a new batch with the same dimensionality as the original input, and a final linear transformation was applied to produce the output.

This setup ensures that both architectures operate sequentially on mini-batches, making the primary difference between them using relational reasoning in the GNN. The results, shown in Figure 15, demonstrate that the GNN-based model outperforms the MLP-based model in this scenario, indicating that the benefits of the GNN arise from its ability to reason over observations within each mini-batch relationally. This experiment highlights the critical role of relational reasoning in achieving better performance.

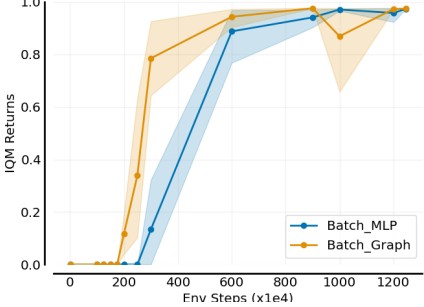

Figure 15: Difference between batch of 50 observations for the Graph and MLP models `MiniGrid-UnlockPickup-v0`

