# OpenReview forum: "Structured Predictive Representations in Reinforcement Learning"
_ICLR.cc/2025/Conference — Submitted to ICLR 2025_

### Official Review · Reviewer_YmHn · 2024-10-28

**Soundness:** 2
**Presentation:** 3
**Contribution:** 2
**Rating:** 3
**Confidence:** 4

**Summary:**

This paper presented a representation learning method in RL, which learns predictable representations by integrating GNNs into a prediction model.  The main motivation is to capture temporal and relational dependencies by constructing graph structures in the latent space. Empirical evaluations are performed on the MiniGrid benchmark. The results show that using GNNs is better than MLPs in terms of performance and robustness.

**Strengths:**

- The paper is well motivated.
- The presentation is good.
- The performed experiment evaluations are relevant.

**Weaknesses:**

The idea of combining GNNs with a forward prediction model for learning predictive embeddings is interesting. But its contribution is limited, as it has been well investigated to learn representations by predicting raw future observations[1] or GNNs[2] in RL.

- The main baseline, min-AIS presented in (Ni et al., 2024), was evaluated in 20 tasks from the MiniGrid benchmarks and several MuJoco tasks. I don't understand why the experiments only consider 4 MiniGrid tasks. I think drawing a conclusion based on the results on the picked 4 tasks is unsuitable.

- According to Figure 4, min-OP is comparable to its counterpart using GNNs on 3 of 4 task, namely DoorKey, KeyCorridorS3R2, UnlockPickup tasks. Therefore, the Graph-based representation learning methods DO NOT outperform the MLP-based methods in all the cases. Therefore the conclusion in line 413 is inaccurate.

- In Figure 5, the interval of min-OP is overlapped with Graph-OP in the presence of distraction changes (a) or size changes (b). So it is wrong to say that Graph-OP outperforms min-OP in ALL scenarios.

[1] Improving sample efficiency in model-free reinforcement learning from images, AAAI 2020.
[2] Contrastive learning of structured world models, ICLR 2020.

**Questions:**

-  I suggest the authors clarify why only the specific 4 tasks are considered. Only these four tasks from the MiniGrid contain subtasks?
-  Providing more evaluations on additional tasks/benchmarks is recommended.
-  Revising the inaccurate conclusions in line 413 and line 437 is suggested.

---

> ### Comment · Reviewer_YmHn · 2024-12-02
> **Official Comments by Reviewer YmHn**
>
> Thank you for your response.
>
> > Comparison with previous work
>
> My main concern is that forward prediction and GNN have been well used for learning representations, limiting the contribution of this work.
>
> > Rationale for selecting the tasks
>
> Selecting 4 tasks from the 20 tasks is a bit cherry-picking. As mentioned, providing more evaluations on additional tasks/benchmarks would be helpful.
>
> Moreover, Figure 4 doesn't show the benefits of using GNN.
>
> I will keep my score.

---

### Official Review · Reviewer_U21z · 2024-10-29

**Soundness:** 1
**Presentation:** 1
**Contribution:** 1
**Rating:** 3
**Confidence:** 4

**Summary:**

The paper introduces a graph neural network based observation-predictive model to learn the latent representation for the downstream reinforcement learning. Experiments on navigation tasks in MiniGrid validate the performance of the proposed algorithm.

**Strengths:**

1. The paper adopts a graph neural network to replace the multi-layer perception to learn the observation representation for a downstream reinforcement learning. Simulation experiments on MiniGrid are presented.
2. The organization is ok.

**Weaknesses:**

1. The paper is largely influenced by the reference Ni et al. (2024), from the problem formulation, method, to experiments. However, some parts inherited from the reference are not related to this paper, which seems irrelevant, as the Q-irrelevant abstractions, model-irrelevant abstraction. The idea is incremental and intuitive. And the contribution compared the reference is minimal.
2. Although the Proposition 3.1 is mentioned in the experimental section, no specific analysis is offered for clear description.
3. Experiments on only one benchmark of MiniGrid is not sufficient, and the performance is also not stably convinced.
4. Beside using the graph neural network as the observation representation learner, transformer is also widely used with better performance, as M-CURL [TPAMI 2023] to learn the state representation, TACO [NeurIPS 2024] to learn the state and action representations, and so on.

There are also some minor writing problems, as
1. “{S2, U, S4, U, S5} (shown in red)” is not consistent with Fig. 2.
2. The same paragraph as the above, the first character of "for instance" should be in capital.
3. Why and how choosing the number of neighbors as m=4?
4. The comparison method as AIS should be clearly stated in this paper to make the paper self contained.

To sum up, the authors are suggested to investigate the literature thoroughly to focus on a more important and clearer problem.

**Questions:**

Will the authors provide more convinced demonstrations with strong baselines to show the significance of the algorithm?

---

### Official Review · Reviewer_jnUY · 2024-11-03

**Soundness:** 3
**Presentation:** 2
**Contribution:** 2
**Rating:** 5
**Confidence:** 4

**Summary:**

The papers discusses predictive state abstractions based on local relationships between trajectories. It shows that incorporating GNNs into the learning process improves the sample efficiency and robustness. The results give further evidence to the known fact that inductive biases can improve learning, if applicable to a certain class of problems such as 2D grid worlds.

**Strengths:**

The paper is written in a reader-friendly way, and the reported results appear to be technically correct.

The main statements are sufficiently well evidenced are likely to be correct.

The theoretical contribution is interesting and important for the considered type of problems.

The paper covers a good range of related work.

**Weaknesses:**

The main problem is the efficiency of the approach compared to other model-based versions of RL: For square grids a set of multi-resolution look-up table would be similarly useful as a maps (or model) in small or medium size grid-world problems. For larger problems, it needs to be be checked whether the GNN does not introduce any errors that could actually be misleading. In other words, the relation of any training errors of the GNN (or a degree of non-stationarity of the task) and the returns would be needed to reasonably assess the benefits of this particular scheme, beyond the general insight that models are useful in well-described (and stationary) environments.

A related problem is the question of applicability to continuous problems, which could have been considered at least in the theoretical part.

Abbreviations (GNN, MLP) need to be be explained in the abstract, because abstracts are to be readable also by non-specialists. The “O” in POMDP  is usually “observable” (rather than “observed”).

The role of discount factor in Eq. 1 needs to be reconsidered or explained.

Sect. 2.2: Put a full stop before “This”.

The paper is not particularly clear about the details of the experiments, although these can be inferred from other papers. E.g.: “We periodically increase the size”: If the size is always increased, it is not strictly “periodical”. Also, it is necessary to state the size range (8x8 and above?). To what level this range is explored? Is the size change even visible in the observations? Likewise, the number of "distractors" is only indirectly stated and whether there are particular arrangements of distractors and how they are affecting the agents perception or action, remains open. Also, changes of topology are mentioned, but there is no further information about this.

**Questions:**

What are the details of the experiments?

Couldn't the algorithm be tested on a problem where size is larger and the topology is more complex?

Is the approach generalizable to continuous problems?

**Details Of Ethics Concerns:**

I don't think "dual submission" belongs here, because, if I wanted to check this, I would most likely destroy the anonymity of the authors which is prohibited by the code of conduct. However, if I were to accept this option as applicable here, I would need to flag it, because it is *possible* that the paper was submitted also elsewhere, just like I would flag a paper where it is *possible* that any participants were harmed. In other words, this should be checked at a later stage of the review process, if at all.

---

> ### Author Response · Authors · 2024-11-21
> **Author Response**
>
> Thank you very much for your constructive feedback. We are pleased that you found our problem statement interesting and important and that you consider our work sound.
> 1. **Clarity in Abbreviations and Terminology:** We have clarified the abbreviations (e.g., GNN, MLP) in the abstract and introduction to ensure the paper is accessible to non-specialists. Additionally, we have corrected the terminology in "POMDP," replacing "observed" with the standard "observable."
> 2. **Multi-Resolution Maps in RL:** Could you please clarify your comment on "multi-resolution look-up tables"? To the best of our knowledge, no work has explored this concept in the context of Minigrid or similar RL settings. If you have any references or examples in mind, we would greatly appreciate your guidance in understanding how this concept could apply to our problem setting.
> 3. **Accuracy of the GNN:** To address your point about the accuracy of the GNN, we are reconstructing the observations created by the GNN and juxtaposing them with raw observations. These results will be included as appendices in the updated manuscript by the end of the review period. We believe this will provide further clarity on the accuracy and robustness of the GNN.
> 4. **Experimental Details:** We acknowledge that specific experimental details were unclear. We have added a detailed table to address this in the updated manuscript. This table includes the range of grid sizes tested, the number and arrangement of distractors, the specifics of topology changes, and their impact on observations. We hope these clarifications address your concerns regarding the clarity of our experiments.
> 5. **Applicability to Continuous Problems:** While our latent state representations are continuous, we currently append actions as one-hot encodings. A straightforward way to extend this to continuous settings would be to incorporate parameterized embeddings for actions. While we have yet to evaluate our approach in fully continuous environments, prior work using GNNs in continuous domains suggests they can capture relational dependencies in these settings. As such, our method will generalize effectively to continuous control problems with minimal modifications. We have included this as an important avenue for future research in the revised paper. We plan to validate this in future work by extending the approach to benchmarks such as Robotic Manipulation environments.
> 6. **Ethical Concern:** We would like to clarify that an earlier version was accepted at a non-archival workshop. It has not been submitted to any archival venue. We hope this alleviates your concern regarding dual submission.

---

> > ### Comment · Area_Chair_Qj5f · 2024-11-24
> > **Please respond to rebuttal ASAP**
> >
> > Dear reviewer,
> > The process only works if we engage in discussion. Can you please respond to the rebuttal provided by the authors ASAP?

---

### Official Review · Reviewer_KsLw · 2024-11-03

**Soundness:** 3
**Presentation:** 3
**Contribution:** 2
**Rating:** 5
**Confidence:** 3

**Summary:**

The paper introduces Message Passing Graph Neural Networks as network architecture in self-predictive RL.

**Strengths:**

- The paper is clear and well-written
- The paper addresses a compelling problem for the community.
- The idea is well motivated

**Weaknesses:**

Flaws in Notation/Clarity:
- minor notational mistakes (for instance Equations (1), (RP))
- General typos (L189, 228, 292, 294, 330+331,…)
- Usually, “value based methods” refer to methods, that don’t learn a policy (L.119)
- Term “Q-Function” is not explained
- subscript of h seems to have different meanings (L. 217 & L. 124)


Method:
- Proposition 3.1 is not clear, as not all spaces and metrics are formally defined. Moreover, there appears to be no proof for this proposition.
- It is not clear how this Theorem provides the foundation for the hypothesis “Training a model to minimize the loss L by reasoning over both these trajectories ensures that the model generalizes between these subtasks, capturing the similarities between these histories.”, especially how this generalises between subtasks.
- It is not clear how the construction of the graphs (L 274 -279) enforces the intuition behind the example in Figure 2.


Experiments:
- The results on some environments look substantially worse than the results presented in the baseline paper (Comparing Figure 4 here with Figure 6 in (Ni et al., 2024)).
- I assume that the shaded regions in figure 4 and 5 respectively are 95%-CIs. Due to (partially) high overlap, it is not so clear, that the Graph Neural Network based method outperform the MLP counterpart.

**Questions:**

- Were different sets of hyperparameters chosen for the different network architectures? (Graph_OP and min-OP for example)? And how do the network sizes compare?
- I am wondering, if only providing Learning Curves as metric is sufficient to support your hypothesis, that the use of GNNs is beneficial. For instance some structure analysis on the encoders (as in Figure 1) would provide additional insights.
- Could you provide a proof for Proposition 3.1?
- Could you provide additional explanation about the importance of that theorem for your approach?

---

### Official Review · Reviewer_MwLh · 2024-11-03

**Soundness:** 3
**Presentation:** 4
**Contribution:** 2
**Rating:** 8
**Confidence:** 3

**Summary:**

This paper introduces a GNN-based approach to representation learning, specifically geared towards environments with sparse rewards and partial observability, that aims to improve sample efficiency and robustness by taking advantage of spatial and temporal structure in trajectories. Building upon past work on self-predictive methods for representation learning (Ni et al. 2024), the authors demonstrate improvements in selected MiniGrid environments. Additionally, the authors show that representations learned with the proposed method generalize better to distractors or size changes in the environment.

**Strengths:**

Originality:
- The paper investigates a novel question, namely can graph-structures be leveraged when learning representations. A brief search shows the contributions of the paper are indeed original work (I was unable to find other GNN-based representation learning methods for RL).

Quality:
- The quality of the paper is generally quite good, with sufficient motivation and background, before introducing results for the proposed method.
- The experiments demonstrate the authors claims, that learning from structure across a batch of latent states has improved sample efficiency and increased robustness. They demonstrate good experimental design, with the only changed factor being the GNN operating over trajectories instead of the MLP operating on a single state in the batch.

Clarity:
- I think the paper is very well-written and clear in its presentation. Specifically, sections 2.1, 2.2 and 3.2 are particularly excellent.

Significance:
- The significance of the paper is moderate. The contributions demonstrate some improvement, showing benefit over prior work, via a new proposed addition. However, there could be more experimentation to demonstrate how this method performs in different, more complex environments and gain further insight into the structures learned.

Overall, the authors present a novel, yet slight, GNN-based modification to an existing self-predictive representation learning method, with the aim to improve the quality of learned representation by leveraging spatial and temporal structure across trajectories. The quality, novelty and clarity of the paper and it's contributions are solid. I would be willing to increase my score if the authors were able to demonstrate or justify why this method would perform well in environments more complex than gridworld-based environments, where the inherent structure the method claims to take advantage of would be far more complex. Additionally, I would be able to increase my score for the significance of this paper if there were more results on the insights gleaned from the structures this method learns. Lastly, I also have some questions around experimental results and baseline design, for which I would appreciate clarifying responses from the authors.

**Weaknesses:**

There's a few areas of improvements that are warranted by the paper.
- While the overall clarity is great, it was not immediately clear in the abstract and introduction that the authors were focused on environments with partial observability (and hence the selection of MiniGrid as their testbed). It's clear that as you read the paper that their work focuses on the POMDP model (history encoder, extending the objectives to the POMDP case, etc.). Additional clarity would make this paper slightly stronger.
    - Specifically, line 17, 53, etc. says in RL, but should be more specific.
- The work only tests their method on a gridworld domain (MiniGrid) and it's not clear if this work can generalize to more complex environments, where perhaps the graph structure is more complicated and similarities between trajectories are less clear.
- Further experiments to understand what the learned structure in the latent space would be very useful to provide further insight into _why_ the GNN-based method works in various settings. Work similar to Figure 1 for other environments enhance the contributions of the paper. Such results would bolster the intuition described in Section 3.1. It would also help strengthen the significance of the paper and it's contributions, because it would shed light on how such structures can apply to other
- The baselines comparison could be substantially stronger. The authors only compare to a version their method without the GNN, but not against other variants or other methods in the literature (such as the ones compared to in Ni et al. or other cited work). Concretely, the authors say "the MLP predicts the next observation for each latent state in a batch and does not use relational reasoning for the whole batch." I understand this to mean that the MLP predicts each state in the batch independently. However, an ablation study where the authors compare to a variant that is an MLP allowed to predict over the entire latent batch, like the GNN, but with fully connected layers instead of message passing, would be helpful in isolating what element of the method makes the largest contribution (or if you need both).
    - Also, if I misunderstood the MLP baseline, and it is actually a fully connected layer that operates over the states in a batch (i.e. is provided the trajectories instead of individual states), then I would make the same argument about comparing to the single-state version. More baselines would be helpful to fully understand the nature of the improvement.

Nits (did not affect score but just flagging for the authors):
- There's a grammatical error in the sentence construction on line 155.
- There's a typo on line 291, extra observation.
- Line 423 has a typo on `tMiniGrid`, should be `MiniGrid`.
- Line 537 should be `incorporate more algorithms.` (`into it` is not quite grammatical)

**Questions:**

Questions:
- is min_OP the same as OP in the (Ni et al. 2024) paper? If so, why does the min_OP curve on various environments not meet the same score in the same samples as the Ni et al. paper? For example, DoorKey-8x8-v0 in the original work achieves close to 0.9 in around 0.5 million steps, but this paper shows 0.7 return in 3 million steps. (It's possible this is because the original work reports episode return, but the authors report IQM -- but I'm slightly concerned about the difference in results).
- Could you clarify what exactly is the input for the MLP baselines? Are they provided trajectories and it's simply a fully connected layer instead of message passing? Or are they provided single states and it's not actually over a trajectory. The confusion arises because it's not clear what you mean by relational reasoning. Additional clarification and detail here would be helpful, beyond just this one sentence description.
> The MLP predicts the next observation for each latent state in a batch and does not use relational
reasoning for the whole batch.

---

### Meta-Review · Area_Chair_Qj5f · 2024-12-20

**Metareview:**

This work studies how incorporating a graph structured prediction into an observation prediction learning process can help improve sample efficiency and robustness. The key method insight is to take a latent self predictive model and then incorporate a GNN based prediction module into this. This is shown to help in robustness and efficiency over some minigrid experiments.

Strengths
The idea makes intuitive sense and seems useful across many domains
There is theoretical backing to the work

Weaknesses
The contribution over Ni et al does seem somewhat minimal and incremental
The experimental results are only on a single, relatively simple domain. Given the incremental nature of the methodology, much greater empirical evidence is needed to support the claims being made in the paper.

Overall I think the paper has merit, but for it to reach it's full potential needs to be tried across a variety of more complex environments, with continuous spaces and so on. This will make a resubmission much stronger!

**Additional Comments On Reviewer Discussion:**

The reviewers brought up concerns about incremental novelty, simple experiments, better baselines, additional visualizations and latent structure analysis. The authors did address some of these concernns, but my primary concern about the relatively limited empirical evaluation remains. Improving this would strengthen the paper overall.

---

### Decision · Program_Chairs · 2025-01-22

Reject